



# The Ant-Iso dataset: a compilation of Antarctic surface snow isotopic observations

Jiajia Wang[1], Hongxi Pang[1], Shuangye Wu[2], Spruce W. Schoenemann[3], Ryu Uemura[4], Alexey Ekaykin[5,6], Martin Werner[7], Alexandre Cauquoin[8], Sentia Goursaud Oger[9,*], Summer Rupper[10], and Shugui Hou[1,11]

[1] Key Laboratory of Coast and Island development of Ministry of Education, School of Geography and Ocean Science, Nanjing University, Nanjing, China

[2] Department of Geology and Environmental Geosciences, University of Dayton, Dayton, USA

[3] Environmental Sciences Department, The University of Montana Western, Dillon, MT, 59725, USA

[4] Graduate School of Environmental Studies, Nagoya University, Furo-cho, Chikusa-ku, Nagoya, 464-8601, Japan

[5] Climate and Environmental Research Laboratory, Arctic and Antarctic Research Institute, 38 Beringa st., 199397 St. Petersburg, Russia

[6] St. Petersburg State University, 33–35, 10th line VO, 199178 St. Petersburg, Russia

[7] Alfred Wegener Institute, Helmholtz Centre for Polar and Marine Sciences, Bremerhaven, Germany

[8] Institute of Industrial Science (IIS), The University of Tokyo, Kashiwa, Japan

[9] Department of Earth Sciences, University of Cambridge, Cambridge, UK

[10] Department of Geography, University of Utah, Salt Lake City, UA, United States

[11] School of Oceanography, Shanghai Jiao Tong University, Shanghai, China

[*] Currently CEA, DAM, DIF, F-91297 Arpajon, France

Correspondence to: Hongxi Pang (email: hxpang@nju.edu.cn) and Shugui Hou (email: shuguihou@sjtu.edu.cn)

**Abstract.** Stable water isotopic observations in surface snow over Antarctica provide a foundation for validating isotopic models and interpreting Antarctic ice core records. Here, we present a new compilation of Antarctic surface snow isotopic



dataset with strict quality control from published and unpublished sources including
measurements from snow pits, snow cores, ice cores, deep surface snow, and
precipitation (multi-year average values). The dataset contains a total of 1867 data
points, including 1604 locations for oxygen isotope ratio ($\delta^{18}O$) and 1278 locations
for deuterium isotope ratio ($\delta^2H$). 1204 locations have both $\delta^{18}O$ and $\delta^2H$, from which
d-excess (d-excess = $\delta^2H - 8 \times \delta^{18}O$) can be calculated. The dataset also contains
geographic and climate information. The database has a wide range of potential
applications, such as the study of the spatial distribution of water isotopes in
Antarctica, the evaluation of climate models, and the reconstruction and interpretation
of Antarctic ice core records. As an example of model evaluation, the compiled
isotopic dataset is used to assess the performance of isotope-enabled atmospheric
general circulation models (AGCMs) on simulating the spatial distribution of water
isotopes over Antarctica. This dataset is the most comprehensive compilation so far of
observed water isotope records at multi-year average scale from multiple sources for
Antarctica. It is available for download at https://doi.org/10.5281/zenodo.7294183
(Wang et al., 2022).

## 1 Introduction

Under the current global warming, changes in the Antarctic continent and ice sheet

have profound impacts on the global sea level, the atmospheric circulation, and many
other important aspects of the Earth systems (Medley and Thomas, 2019; Naughten et
al., 2021; Stokes et al., 2022). To better understand and predict climate change in
Antarctica, it is important to investigate its past variations, which is limited by the
length of instrumental data (reanalysis data and automatic weather station), covering
only about 60 years. Therefore, assessing long-term climate change in Antarctica
requires climate proxy records. Ice cores from Antarctica are critical archives for
climate change due to their high resolution, long temporal coverage, large amount of
information, and high fidelity. As one of the most important ice core records, the
stable water isotopic record is used to reconstruct past temperatures and provides a
fundamental understanding on past climate change (Brook and Buizert, 2018; Buizert
et al., 2021; Jouzel et al., 2007).

Conventional temperature reconstructions by stable water isotopic records in ice

cores rely on the empirical spatial linear relationship between the isotopic
composition in surface snow and air temperature. The establishment of such
relationship requires sufficient observational data. Despite the large number of
isotopic observations made in Antarctica over the past few decades, the spatial
coverage is still uneven. Masson-Delmotte et al. (2008, hereafter MD08) compiled the
first multi-year-averaged Antarctic surface snow stable isotope dataset, which
provided a solid foundation for related research on Antarctic water isotope
climatology. First, it provides an observational basis for numeric simulations of the
spatial distribution of snow isotopes across Antarctica using pure mathematical
methods combined with high-resolution digital elevation models (Hatvani et al., 2017;
Wang et al., 2009a, 2009b, 2010). Second, it can be used to reconstruct
paleo-temperature and paleo-elevation changes (Werner et al., 2018). Finally, the
database can be used as a benchmark to evaluate isotope-enabled atmospheric general
circulation models (AGCMs) and Rayleigh distillation isotope models.

Although isotopic observations have been recorded at over one thousand sampling

sites, the spatial coverage of isotope data remains highly uneven (MD08). Only a few
data are available on the West Antarctic Ice Sheet and at high altitudes of the East
Antarctic interior regions. After the pioneering work of MD08, numerous new
samples and measurements have been acquired by different researchers. Incorporating
these additional observations, we have recompiled the most comprehensive Antarctic
surface snow isotopic dataset, which significantly increases the spatial coverage
relative to the MD08. This article aims to provide some details on the collection of the
isotopic measurements used to produce this updated Antarctic surface snow isotopic
dataset, including data sources, data spatial distribution, and data selection criteria. In
addition, we use this dataset to assess the performance of isotope-enable AGCMs as
an example of its potential applications.

## 2 Description of the Antarctic surface snow isotopic dataset


### 2.1 Data collections and sources


The Antarctic surface snow isotopic data were collected from published papers and
public data portals. When the raw data used in the published studies were not publicly
available, we requested the data from the authors. We received strong support in the
process of data collection. Figure 1a shows the data points in the original MD08
dataset. Figure 1b shows our newly added observation points (794 new data points),
and Figure 1c shows the total sampling points (1867 data locations) of our updated
dataset.
Traverse sampling is an important way to obtain spatially distributed data of
isotopic composition. Therefore, the new data mainly came from different route
traverses (Fig. 2). In particular, over the Dronning Maud Land (DML) (from 0 to
60°E), we compiled the water isotope data from the Swedish−Japanese traverse
between Syowa Station and Dome F Station (Touzeau et al., 2016), the Japanese
Antarctic Research traverse (Uemura et al., 2016), the Coldest Firn (CoFi) project
traverse from Kohnen Station to Plateau Station (Weinhart et al., 2021), and the
Spanish expedition from Novolazareskaja Station to South Pole Station (Landais et al.,
2017). Our dataset also included published records from the Chinese traverse from
Zhongshan Station to Dome A, near the Lambert Glacier (Li et al., 2014, 2021). For
the high inland region of East Antarctica (90 to 120°E), we added two routes of
traverse data (Vostok−Dome B and Vostok−Dome C traverses). Furthermore, in the
area around the Vostok site, we included 89 unreleased public snow pits data, such as
the Vostok flow line data (Ekaykin et al., 2012). Previous data over the West
Antarctica Ice Sheet (WAIS) area mainly came from the International
Trans-Antarctica Scientific Expeditions (Qin et al., 1994; Steig et al., 2005). The
updated dataset includes more recent traverses of the WAIS by Brazilian and Chilean
researchers from the Möller Ice Stream (MIS) basin to the Pine Island Glacier (PIG)
basin (Marcher et al., 2022), and from Patriot Hills to South Pole (Marquetto et al.,
2015). It also includes the Satellite Era Accumulation Traverse (SEAT) firn core data
collected during the 2010−2012 field season on the WAIS ice divide (Burgener et al.,
2013; Williams, 2013).



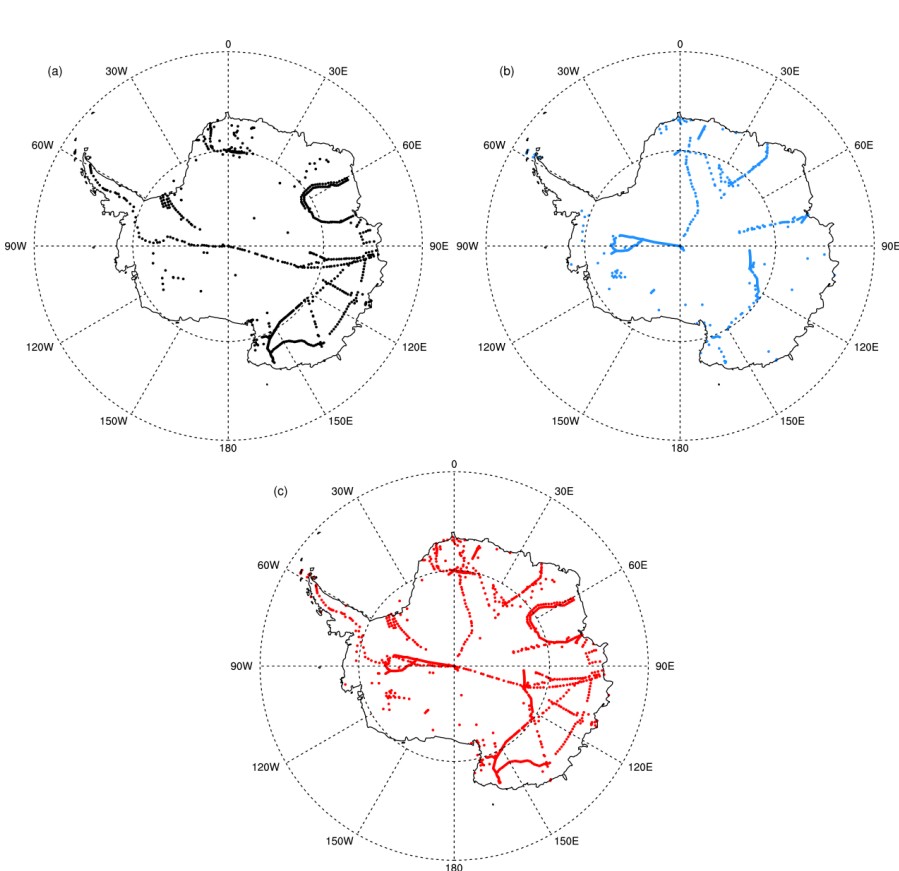


Figure 1. The comprehensive dataset of Antarctic surface snow isotopic observations.
The black points indicate the original dataset of MD08 (a), and the blue points
represent our newly added points (b), and the red points indicate all data locations (c).






Figure 2. Major new sampling traverses of isotopic observations in Antarctica.


## 2.2 Selection criteria and types of data collected in the Antarctic surface snow isotopic dataset

We applied strict data selection criteria to ensure consistent quality over the entire dataset. As we aimed for a reliable isotopic dataset of multi-year averages, we excluded isotopic data with apparent seasonal bias and those modified by post-depositional processes. For example, previous studies reported a high number of surface snow samples with negative d-excess values in the Dry Valleys (Gooseff et al., 2006; MD08; Hu et al., 2022). They likely came from fresh surface snow or surface glacial ice, which could not represent the multi-year average isotope values. Therefore, we excluded these surface samples from the Dry Valleys (Gooseff et al., 2006; MD08), and only retained data from snow pits. For snow pits and shallow firn ice



cores, data should contain at least one-year's record. For longer ice cores, only data of
the last few decades were included. For precipitation, data should cover at least one
year. For the surface snow, samples should have a thickness adequate to cover a full
year's accumulation. For example, surface snow samples from East Antarctic coastal
sites should have a depth of at least 30 cm, while those from central West Antarctic
Ice Sheet should have a depth of at least 15−20 cm. Data were averaged at each site to
get the multi-year mean.
**2.3 Metadata**
Table 1 provides a detailed description of metadata fields used in our Antarctic
surface snow isotopic observation dataset. Key information included the site latitude
and longitude, geographic factors (elevation and distance to the nearest coast), climate
conditions (temperature and net snow accumulation rate), and isotopic values.
Latitude and longitude data were provided in the literature or by contributing
researchers. Elevations were extracted from the Global Earth Relief Grids data
(https://lpdaac.usgs.gov/products/srtmgl3v003/) at a spatial resolution of 0.09 km. We
applied the built-in functions in the Generic Mapping Tools (GMT) software (Wessel
et al., 2019) to calculate distance from the nearest coast. Firn temperature or surface
air temperature data were obtained either from the literature or provided by the
contributing researchers. The net snow accumulation rate, when available, was
expressed in cm w.e. yr$^{-1}$. The dataset contains 1867 data points in total, including
1604 locations for $\delta^{18}O$ and 1278 locations for $\delta^{2}H$, and 1204 positions for both,
where the d-excess (defined as d-excess = $\delta^{2}H − 8 \times \delta^{18}O$) could be calculated.

Table 1. Descriptions of metadata fields in the Antarctic surface snow isotopic
observation dataset



| Field number | Information |
| --- | --- |
| 1 | Sample ID number |
| 2 | latitude (decimal degrees) |
| 3 | longitude (decimal degrees) |
| 4 | site name |
| 5 | elevation (m) |
| 6 | sampling date |
| 7 | sample type |
| 8 | reference or source |
| 9 | published distance from the nearest coast (km) |
| 10 | calculated distance from the nearest coast (km) |
| 11 | firn temperature or surface air temperature (°C) |
| 12 | accumulation of snow/ice per year (cm w.e./yr) |
| 13 | averaging length (years or depth) |
| 14 | Number of averaged values |
| 15 | mean $\delta^2$H (traditionally referred to as $\delta$D, ‰) |
| 16 | min $\delta^2$H (‰) |
| 17 | max $\delta^2$H (‰) |
| 18 | $\delta^2$H standard deviation (‰) |
| 19 | mean $\delta^{18}$O (‰) |
| 20 | min $\delta^{18}$O (‰) |
| 21 | max $\delta^{18}$O (‰) |
| 22 | $\delta^{18}$O standard deviation (‰) |
| 23 | mean d-excess (‰) |



| 24 | max d-excess (‰) |
| --- | --- |
| 25 | min d-excess (‰) |
| 26 | d-excess standard deviation (‰) |
| 27 | calculated elevation (m) |
| 28 | place of measurements (country) |
| 29 | original quality control: (1) analytical uncertainty of 0.1‰ or better for $\delta^{18}O$ measurements, (2) analytical uncertainty of 1.0‰ or better for $\delta^2H$ measurements, (3) sufficient number of measurements (10 or more), (4) age control on the sampling period or a core depth), (5) seasonal resolution of the measurements |

Note: The original quality control index ranging from 0 (minimum quality control) to
5 (maximum quality control).

## 3 Annual mean spatial distribution of Antarctic $\delta^{18}O$ and d-excess and evaluation of isotope-incorporated AGCMs



The measured $\delta^{18}O$ values range from –59.95 to –7.80‰, with the maximum and
minimum values at the Antarctic Peninsula and near Vostok Station respectively (Fig.
3a). The d-excess values range from –5.8 to 21.8‰, with the minimum and maximum
values in the Dry Valleys and near Vostok Station respectively (Fig. 3b). The spatial
pattern clearly shows the continental effect with the $\delta^{18}O$ values decreasing from the





coast to the interior regions due to the temperature-dependent isotopic distillation (Fig.
4a, 4c, and 4e), and d-excess increasing from the coast to the interior (Fig. 4b, 4d, and
4f) because of the equilibrium and kinetic fractionation effects occurring during the
formation of ice crystals at very low temperatures (Jouzel and Merlivat, 1984). It
should be noted that we do not quantitatively calculate the quantitative relationship
between isotope ratios and geographical and climatic factors here, which is beyond
the scope of this paper.

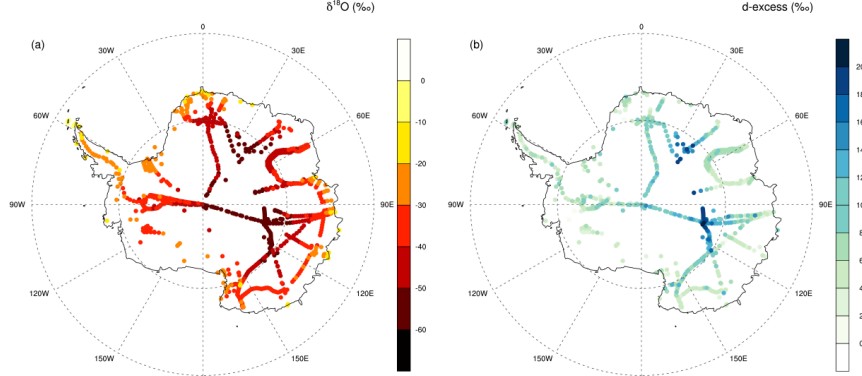


Figure 3. Spatial distribution of (a) $\delta^{18}O$ and (b) d-excess in Antarctic surface snow.






Figure 4. Scatter plots for (a) δ¹⁸O vs. distance to the nearest coast and (b) δ¹⁸O vs.
elevation. (c) and (d), same as (a) and (b), but for d-excess. (e) Scatter plot for δ¹⁸O vs.

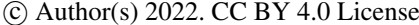

temperature. (f), same as (e), but for d-excess. The open circles indicate the original
data set of MD08, and the filled points represent our newly added points.

In order to compare our isotopic dataset with the simulations from
isotope-incorporated AGCMs, we used the published results of two recent versions of
isotope-enabled AGCMs: the European Centre for Medium-Range Weather
Forecasting-Hamburg Atmosphere Model equipped with water isotopes, version 6
(ECHAM6-wiso) and the isotope-enabled Community Atmosphere Model, version 6
(iCAM6). The ECHAM6-wiso model was driven by the ERA5 reanalysis dataset
(Hersbach et al., 2020) with a horizontal resolution of 0.9° x 0.9° for the period 1979
to 2018 (Cauquoin and Werner, 2021). The iCAM6 model was also driven by the
ERA5 and the simulation time spans from 1980 to 2004 with a horizontal resolution
of 0.9° x 1.25° (Fiorella et al., 2021). These two recent versions of isotope-enabled
AGCMs could well reproduce the spatial distribution characteristics of Antarctic
precipitation $\delta^{18}O$ and d-excess (Fig. 5). The correlation coefficient between
measured and observed $\delta^{18}O$ is slightly higher for ECHAM6-wiso (r = 0.97,
model-data slope = 0.78, p < 0.01) than iCAM6 (r = 0.96, model-data slope = 0.84, p
< 0.01) (Figs. 5a and 5b, Figs. 5e and 5f). The performance of ECHAM6-wiso in
simulating d-excess (r = 0.77, model-data slope = 0.22, p < 0.01) is significantly
inferior to iCAM6 (r = 0.89, model-data slope = 1.01, p < 0.01) (Figs. 5c and 5d, Figs.
5g and 5h). The iCAM6 was improved over its predecessor iCAM5 in several
significant ways, particularly in its cloud parameterizations (e.g., Bogenschutz et al.,
2018). These changes included a revision to the contact angle distributions (Wang et
al., 2014), how pre-existing ice crystals influence ice nucleation rates (Shi et al.,
2015), and a new prognostic microphysics scheme (Gettelman et al., 2015). Together,

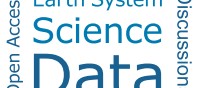

these changes allowed for lower supersaturation, which led to an increase of the
modeled d-excess in Antarctica by 5–10‰ (see Figure 4 of Fiorella et al. (2021)).
This could explain the iCAM6's better ability to model d-excess of snow and
precipitation in Antarctica in comparison to ECHAM6-wiso. ECHAM6-wiso, when
nudged to the ERA5 reanalyses, significantly underestimated the d-excess in
Antarctic precipitation. This bias was slightly reduced when using the results from a
preindustrial simulation instead (average climatic conditions from 1870 to 1899) (r =
0.84, model-data slope = 0.31, p < 0.01, see Supplement Fig. S1). This simulation was
used by Cauquoin and Werner (2021) to re-tune the supersaturation equation used in
ECHAM6-wiso, based on the reproduction of the observed d-excess/$\delta^2$H relationship
from the Antarctic surface snow isotopic dataset of MD08. In conclusion, two recent
versions of isotope-enabled AGCMs were able to reproduce the isotopic composition
of precipitation and snow in Antarctica. This information could give us more
confidence in using these water isotope simulations in polar regions for paleoclimate
reconstruction.








Figure 5. Comparisons of observational water isotopes data with the ECHAM6-wiso
and iCAM6 simulations. (a) The observational $\delta^{18}$O (filled circles) and the simulated
$\delta^{18}$O (background color ramp) by ECHAM6-wiso model. (b) Scatter plot of the
simulated $\delta^{18}$O (y-axis) vs. observational $\delta^{18}$O (x-axis). (c) and (d), same as (a) and
(b), but for d-excess. (e) and (f), same as (a) and (b), but for iCAM6. (g) and (h), same
as (e) and (f), but for d-excess. Letter r represents the correlation coefficient, slope
represents the slope of linear regression, and rmse represents the root mean square
error.
**4 Data availability**
The updated Antarctic surface snow isotopic dataset used in this article is available
at https://doi.org/10.5281/zenodo.7294183 (Wang et al., 2022).
**5 Conclusions and outlook**
We compiled a multi-year-averaged Antarctic surface snow isotopic dataset by
integrating a previous database with more recent observations. This dataset greatly
improved the spatial coverage of isotopic observations in Antarctica. As an example
of its potential applications, we made a comparison between the isotopic observation
data and the simulation results from the two most recent isotope-enabled AGCMs.
The results show that the ECHAM6-wiso and iCAM6 in general captured the spatial
variation and characteristics of Antarctic precipitation isotopes. The updated dataset
has many important potential applications in investigating spatial variability and the
climatology of water isotopes in the region, evaluating models, and interpreting
Antarctic ice core records for past climate variations.
For better data integration and update, we make the following recommendations for
future studies: (1) more data are needed at high elevations in interior Antarctica; (2)
data collection techniques such as dual-tube sampling are needed to ensure annual
data coverage and reduce seasonal biases; (3) all isotopic observation data should be
made publicly available. Finally, we look forward to collaborating with a wide range
of researchers to update and refine this dataset on regular basis.

**Supplement.** GMT programs include DEM data that calculate the distance to the
nearest coast and extracts the site's elevation.
**Author contributions.** HP and SH initiated the project leading to the collection
of the Antarctic surface snow isotopic dataset. JW was responsible for collecting and
collating all the data, as well as completing the first draft of the article. The original
draft was improved by SW, SWS, RU, AE, MW, AC, SGO, and SR. HP and SH
supervised this work. All authors contributed to the final form of the manuscript.
**Competing interests.** The authors declare that they have no conflict of interest.
**Acknowledgements.**
This work was jointly supported by the National Natural Science Foundation of China
(41622605, 41830644, and 91837102), the Priority Academic Program Development
of Jiangsu Higher Education Institutions (PAPD), the Collaborative Innovation Center
of Climate Change of Jiangsu Province of China, the Fundamental Research Funds
for the Central Universities (020914380190), and the ERC Starting Grant
COMBINISO (306045). We thank Jean-Louis Tison for his support and Jesse
Nusbaumer for the interpretation of the iCAM6 results. We are grateful for the help of



Jean Jouzel and Jiandong Xu. Finally, we thank the scientists who have provided help
and support in data collection.

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
