# Peer review of "The Ant-Iso dataset: a compilation of Antarctic surface snow isotopic"

_Earth System Science Data, 2022_

## Author Comment (AC1)

Specific comments:

Why authors of the Antarctic surface snow isotopic dataset available for download at https://doi.org/10.5281/zenodo.7294183 are different from the authors of the submitted ESSD article?

Response: We have updated the authors of the dataset to match the authors of this article (please see https://doi.org/10.5281/zenodo.7788716).

It is not clear why the title of ESSD paper is "The Ant-Iso dataset: a compilation of Antarctic surface snow isotopic observations" while there are 80 points of ice cores and 235 points of firn cores. You should either revise the title or exclude ice cores from the dataset.

Response: We have changed the title. The new title is *"The Ant-Iso dataset: a compilation of Antarctic surface snow and ice isotopic observations"*.

The dataset needs to be revised and polished. You should thoroughly check the dataset to be sure that it follows the common requirements of a dataset. At least 25 samples do not have coordinates, elevation and year of sampling. 117 samples do not have coordinates and year of sampling. I doubt that such values without any spatial and temporal references could be useful. If it is not possible to obtain the metadata, they should be excluded from the dataset. By the way column "Sample label" suggests location for some of the samples without coordinates. For example, 753 Molodezhnaya, 754 Amery-G1, 755 GM7, 756 GM10, 757 GM13, 758 Dome C, 759 Mirny, 760 Pioneerskaya, 761 Vostok 1, etc. Probably you can use it after careful check.

Response: We have deleted the points without latitude and longitude information and added the sampling time information as much as possible. Most of the missing information about sampling was in the original MD08 dataset. Through hard work, the sampling time is more complete in the new dataset. Based on your suggestion, we significantly revised and improved the dataset.

You need to define parameters in the dataset. It is not clear what the difference between published and calculated distance is and why did you need to calculate it? The same relates to elevation.

Response: There are quite a few points with latitude and longitude coordinates, but no information about elevation and distance to the nearest coast. For those data points, we extracted their elevation from the Global Earth Relief Grids data, and calculate distance to the nearest coast using the Generic Mapping Tools (GMT) software (Wessel et al., 2019). Therefore, some elevation and distance to coast data came from the same publications as the isotope data, but some came from our calculations. We added a supplementary explanation in the excel table.

It is also not clear how the quality flag was assessed.

Response: The quality control value ranges from 0 (minimum quality control) to 5 (maximum quality control), which is consistent with the practice of MD08. Data that meet any of the quality control index conditions can get 1 point: (1) analytical uncertainty of 0.1‰ or better for $\delta^{18}O$ measurements, (2) analytical uncertainty of 1.0‰ or better for $\delta^2H$ measurements, (3) sufficient number of measurements (10 or more), (4) age control on the sampling period or a core depth, (5) seasonal resolution of the measurements. Note: The purpose of the first two quality standards is to reflect

the quality of sample preservation and water isotope measurements; the latter three are based on the time scale and resolution of the samples in order to establish a climatic dataset on the isotopic composition of surface snow and ice (MD08).

*"The quality control value ranges from 0 (minimum quality control) to 5 (maximum quality control). Data that meets any of the quality control index conditions (Table 2) can get 1 point."*

*"Note: The purpose of the first two quality standards is to reflect the quality of sample preservation and water isotope measurements; the latter three are based on the time scale and resolution of the samples in order to establish a climatic dataset on the isotopic composition of surface snow and ice (MD08)."*

Does "Firn temperature or surface air temperature" relate to exact location and time of sampling or is it somehow averaged? How was it calculated or assessed? What is "Accumulation of snow/ice per year"and how was it estimated?

Response: "Firn temperature or surface air temperature" is related to the exact location and time of sampling. The air temperature data of some stations are calculated based on the weather stations. But for most sites, the multi-year average temperature data are not available. Samplers usually measure the firn temperature at 10 m of the ice core. According to the thermodynamics of glacier, the 10 m firn temperature can represent the annual average surface air temperature of the sampling site (MD08; Sun et al., 2021).

"Accumulation of snow/ice per year" is the net accumulation of annual precipitation, that is, surface mass balance. In Antarctica, there are five ways to get the accumulation rate (snow pit, ice core, stake observations, ground-penetrating radar

and automatic weather station). Snow pits and ice cores are first dated, and then the accumulation rates are calculated based on the annual layer thickness. The snow accumulation calculation of a site can also be directly measured by stakes, ground-penetrating radar and automatic weather station. The accumulation rate data are also available in the literature.

Data in columns should be formatted in a single style. For example, column "Averaging length (years or depth)" contains different data in very different style that prevent easy processing and analysis of the dataset. I suggest splitting the column into two different ones ("Averaging years" and "Averaging depths") and putting only numerical values in each of them. If needed additional explanation you could add another "Comments" column with text.

Response: Based on your suggestion, we divided this column of data into two columns (Averaging years or year period, Averaging depths).

Column "Sampling date" contains not dates but years in different formats (both numeric and text) that prevent processing and filtering. You may consider splitting the column into two different ones – "sampling year start" and "sampling year finish".

Response: As suggested, we revised all sampling times to make them consistent and easy to use. But it is not practical to have separate columns for start time and end time, because some data only contain a rough sampling range, without exact start and end time.

Column "Sample type" have errors in writing that prevent grouping samples by types. You should carefully check every type and provide exact number of points of every type in the article.

Response: We identified and corrected all the errors in the sample type column, and added their summary statistics in the article. Our dataset includes 885 snow pits, 358 snow cores, 77 ice cores, 359 surface snow samples, and 19 precipitation samples.

*"Our dataset includes 885 snow pits, 358 snow cores, 77 ice cores, 359 surface snow samples, and 19 precipitation samples."*

Lines 68-74. Are you talking about the dataset by Masson-Delmotte et al. (2008), mentioned earlier, or your dataset? Clarify it and if it relates to the dataset by Masson-Delmotte et al. (2008), provide more details about its actual content rather than "it provides an observational basis".

Response: We revised it to "MD08 dataset" to make it clear.

*"MD08 database provides an observational basis for numeric simulations of the spatial distribution of snow and ice isotopes across Antarctica using pure mathematical methods combined with 1 km high-resolution digital elevation models (Wang et al., 2009a, 2009b, 2010)."*

Lines 78-79 Provide sufficient references of "numerous new samples and measurements that have been acquired by different researchers"

Response: We added the references.

*"numerous new samples and measurements have been acquired by different researchers (i.e. Ekaykin et al., 2020; Landais et al., 2017; Weinhart et al., 2021)."*

Lines 79-81 Add quantitative estimation of the "additional observations" and described in numbers the difference between MD08 and your dataset.

Response: As suggested, we added a quantitative estimation: *"The original dataset has 1279 points, and we added 794 new data points."*

Lines 90-91 How many data points did you make publicly available for the first time? It is one of the most important things to show the value of your dataset.

Response: We added 794 new observation points in total. Among them, 226 data points were not available on public repository or supplementary material of papers. We obtained those data by directly contacting the authors (Table 1). We added the following descriptions and tables to the dataset.

*"Among the new additions to the data, 226 data points were not available on public repository or supplementary material of papers. We obtained those data by directly contacting the authors (Table 1)".*

**Table 1.** Data that were not available in public repositories or supplementary materials for literature

| Sampling site | Data numbers | References | Comments |
|---|---|---|---|
| Zhongshan−Dome A | 13 snow pits | Xiao et al. (2013) | Supplemented data via email |
| Vostok flow line | 89 snow pits | Ekaykin et al. (2012) | Supplemented data via email |
| Syowa−Dome F | 51 surface snow samples | Touzeau et al. (2016) | Supplemented data via email |
| SEAT | 14 firn cores | Burgener et al. (2013) and Williams (2013) | Supplemented data via email |

| Zhongshan Station | 1 precipitation sample | This study | In situ collection |
|---|---|---|---|
| Vostok | 21 snow pits | Ekaykin et al. (2016) | Supplemented longitude and latitude information via email |
| South Pole | 10 ice cores | Fudge et al. (2019) | Supplemented longitude and latitude information via email |
| Lambert Glacier | 1 snow pit and 1 ice core | Du et al. (2020) and Liu et al. (2019) | Supplemented data via email |
| DML | 10 firn cores | Schlosser et al. (2014) and Vega et al. (2016) | Supplemented data via email |
| WAIS | 15 ice cores or firn cores | Criscitiello (2014), Tavares et al. (2020), Thomas et al. (2013), Tetzner et al. (2022), Schwanck et al. (2017), Thomas and Bracegirdle (2015) | Supplemented data via email |

Lines 119-121 Add numbers of points to the figure caption

Response: We have changed this figure. Following is the re-drawn Fig. 1.

[Figure]

**Figure 1.** The comprehensive dataset of Antarctic surface snow and ice isotopic observations. The black points indicate the original dataset of MD08 (a), and the blue points represent our newly added points (b), and the black and blue points indicate all data locations (c). Major new sampling traverses of isotopic observations in Antarctica (d).

Lines 118-123 Consider merging figures 1 and 2. You can add information from Fig.2 to Fig.1 (b)

Response: We merged figure 1 and 2 as suggested. See figure 1 above.

Line 177-178 Include the same figure for δD, ‰

Response: We added a new figure for δD. Following is the revised figure.

[Figure]

Figure 2. Spatial distribution of (a) $\delta^{18}O$, (b) δD, and (c) d-excess in Antarctic surface snow and ice.

Line 214 Since you have several files in Supplement you should reference more precisely.

Response: We deleted all but one Word file to avoid confusion.

*"The supplement related to this manuscript can be found in the supplement.docx Word."*

Line 252 Content of the Word file with Supplement differs from the supplement described here. You should provide detailed description of the supplement files.

Response: We deleted all but one Word file to avoid confusion.

*"The supplement related to this manuscript can be found in the supplement.docx Word."*

**Technical corrections:**

Affiliations and even country names have different formats

Response: They were revised in a consistent format.

[1] School of Geography and Ocean Science, Nanjing University, Nanjing, China

[2] Department of Geology and Environmental Geosciences, University of Dayton, Dayton, USA

[3] Environmental Sciences Department, The University of Montana Western, Dillon, MT, USA

[4] Graduate School of Environmental Studies, Nagoya University, Nagoya, Japan

[5] Arctic and Antarctic Research Institute, 38 Beringa St., 199397 St Petersburg, Russia

[6] Institute of Earth Sciences of Saint Petersburg State University, 31–33 10th line V.O., 199178 St Petersburg, Russia

[7] Alfred Wegener Institute, Helmholtz Centre for Polar and Marine Sciences, Bremerhaven, Germany

[8] Institute of Industrial Science, The University of Tokyo, Kashiwa, Japan

[9] Department of Earth Sciences, University of Cambridge, Cambridge, UK

[10] Department of Geography, University of Utah, Salt Lake City, USA

[11] School of Oceanography, Shanghai Jiao Tong University, Shanghai, China

[*] Currently CEA, DAM, DIF, F-91297 Arpajon, France

Line 174 Rewrite "…we do not quantitatively calculate the quantitative relationship…"

Response: This sentence was revised as "*It should be noted that we did not examine the linear correlation between isotope ratios and geographical and climatic factors here, as this is beyond the scope of this paper*."

---

## Author Comment (AC2)

In the manuscript "The Ant-Iso dataset: a compilation of Antarctic surface snow isotopic observations", Wang et al. present a compilation of published and unpublished isotopic data (delta 18O, delta 2H and D-excess). The manuscript is largely well-written and follows a logical structure, but different points were unclear. For instance, it was explained that the "MD08" isotope data was supplemented with 794 newly collected data (L89-93). However, it was not explained:

- which methods were used to find the different studies
- which keywords were used
- which time scales were considered
- why were certain studies in or while maybe others excluded
- why specific water sources were included or excluded
- sampling method
- which data quality parameters were used to decide to in- or exclude
- why certain meta data was presented while others not
- which data was collected by the authors and which from literature (in database)

which methods were used to find the different studies

which keywords were used

Response: Here we answer the two questions together. We have added the following sentence to explain the method and keywords. *"We started the data collection process by searching for scientific journals on Google Scholar with these keywords: Antarctic, surface snow and ice, and water isotope."*

which time scales were considered

Response: Our data has only a time scale, the annual average scale. So we think the time scale you are talking about in this question refers to the time spans. Since we aim to establish the isotopic composition of Antarctic surface snow and ice under modern climatic conditions, we limited our data collection within the past 100 years or so. For

longer ice cores, we only used data for the last few decades. The time period for the majority of our data (~94.2%) is about 1905−2020. Only a few data points in the original dataset have longer timespan. For example, data at Dome B, Vostok, and EPICA Dome C covered about 200 years. We decided to keep these data points given the importance of the Antarctic deep ice core data and the fact that these data can also represent the multi-year average. In revision, we added the timespan of each site whenever such information was available. In columns 13 and 14 in the dataset, we provided the timespan and sampling depth at each site.

*"After a thorough cleaning of the original data, we imposed the following requirements for the newly added data. For snow pits and shallow firn ice cores, data should contain at least one-year's record. Most data came from recent samples under modern climate conditions. Therefore, for longer ice cores, only data of the last few decades were included with the purpose of establishing average isotopic composition of Antarctic surface snow and ice under modern climatic conditions (MD08). For precipitation, data should cover at least one year. For the surface snow, samples should have a thickness adequate to cover a full year's accumulation. For example, surface snow samples from East Antarctic coastal sites should have a depth of at least 30 cm, while those from central West Antarctic Ice Sheet should have a depth of at least 15−20 cm. Data were averaged at each site over time and depth to get the multi-year mean. The majority of our data came from a period between 1905 and 2020 (Ekaykin et al., 2020; Naik et al., 2010)."*

why were certain studies in or while maybe others excluded (1)

why specific water sources were included or excluded (2)

which data quality parameters were used to decide to in- or exclude (3)

Response: These three questions are similar, and we answer them together here.

(1) We noticed that some of the data in the original dataset have obvious flaws and may not represent the annual or multi-year average values. For example, previous studies reported a high number of surface snow samples with negative d-excess values in the Dry Valleys (Gooseff et al., 2006; Hu et al., 2022; MD08). They likely came from fresh surface snow (~5 cm) or surface glacial ice, which could not represent the multi-year average isotope values. Therefore, we excluded these surface samples from the Dry Valleys (Gooseff et al., 2006; MD08), and only retained data from snow pits.

(2) Our core standard is to include only annual average data and exclude and seasonal data. Based on this, we excluded surface snow samples (10 cm) cover from Zhongshan Station to Dome A (Xiao et al., 2013; Pang et al., 2015), and fresh surface (~2 cm) snow along the inland traverse route of Japanese Antarctic Research Expeditions (Horiuchi et al., 2022). Water sources used in this study include snow pits, snow cores, ice cores, deep surface snow, and precipitation (multi-year average), because these sources can represent multi-year average values.

(3) We carefully exclude any samples that did not reflect averages over one year or more. This is a strict standard for our data selection to ensure consistent quality throughout the dataset.

*"Samples that did not reflect averages over a period of one year or longer were excluded from the dataset. As we aimed to compile a comprehensive, reasonable, and quality-controlled isotopic dataset representing multi-year averages under modern climatic conditions, we excluded isotopic data with apparent seasonal biases (such as fresh surface snow) and those modified by post-depositional processes (such as surface glacial ice) at Dry Valleys. Previous studies reported a high number of fresh surface snow and surface glacial samples with abnormal negative d-excess values*

*(the range is about –10 to –5‰) in the Dry Valleys (Gooseff et al., 2006; Hu et al., 2022; MD08). Considering accumulation rates and the influences of post-deposition processes, we believe that it is unlikely that these fresh surface snow (5 cm) and surface glacial ice samples represented the multi-year average isotope values. Therefore, we excluded these surface samples from the Dry Valleys (Gooseff et al., 2006; MD08), and only retained data from snow pits (1 m)."*

sampling method

Response: There are two main ways to obtain snow and ice samples in Antarctica: (1) use rain bucket to collect precipitation or snowfall; (2) use metal instruments to collect surface snow samples and dig/drill snow pits and ice cores. One column was added in the dataset to describe the sampling methods. The following description was added to the manuscript.

*"There are two main methods to obtain snow and ice samples in Antarctica: (1) use rain bucket to collect direct precipitation; (2) use instruments to collect samples from surface snow, snow pits and ice cores."*

why certain meta data was presented while others not

Response: The purpose of our paper is to collect isotope data, so we mainly show the spatial distribution characteristics of isotope data. The spatial distribution characteristics of isotopes are mainly related to altitude, distance to the nearest coast, and temperature, so we didn't show the relationship between isotopes and meta data accumulation rates (Accumulation of snow/ice per year). We have added the spatial pattern of δD in surface snow and ice over the Antarctica. But we did not show the relationships between δD and geographical and climatic factors. The spatial variation

of deuterium and its relationship with geographical and climatic factors are basically the same as oxygen isotopes. Therefore, we did not show the relationships between δD and geographical and climatic factors.

which data was collected by the authors and which from literature (in database)

Response: All data was collected from literature. Samples with ID numbers 1−904 were from the original dataset (MD08), and samples with ID numbers 905 to 1698 were collected from additional published studies by us. The following description was added in the manuscript.

*"In the dataset, samples 1−904 came from the original dataset, and samples 905 to 1698 were newly added."*

In addition, it was not explained (database and text) in which way the different spatial and temporal data, and the model time scale were aggregated.

Response: As for the measured sampling data, the averaging years or year period and averaging depths at each site were added to the database (Columns 13 and 14). The measured data (water isotopes and climatic metadata) were averaged at each site over time or depth to get the multi-year mean values. In order to compare the observed and modeled data, we averaged the model data over time. We then interpolate the model data to the sampling site using linear interpolation method. Next, we performed regression analysis between the measured and modeled data. Although this comparison has some shortcomings, it is reasonable in the average state of climate. We have explained how different spatial and temporal data and the time scale of the model are aggregated in the revised manuscript. The following descriptions were added in the manuscript.

*"Data were averaged at each site over time and depth to get the multi-year mean."*

*"In order to compare the observed and modeled data, we averaged the model data over time. We then interpolate the model data to the sampling site using linear interpolation method. Next, we performed regression analysis between the measured and modeled data. For more details on model evaluation methods, please refer to the supplementary material of Werner et al. (2018)."*

It seems that in the presented data (doi.org/10.5281/zenodo.7294183), figures, and tables, all data was shown at once without distinguishing between different spatial resolutions (space and vertical resolutions), temporal resolutions (years and time single vs time series), and various water sources. It would be good to present all "raw" collected data from literature in the database and explained in the manuscript how the data was averaged (e.g. Pang et al. 2019, snow pit 3 m 29 average values; Münch et al. 2017, snow pit 3.4 m 1329 average values 2014-2015). In addition, discuss the value of such averaged data and whether the collected data can be used to compare field data and model results consisting of different time scales. Providing the full collected data allows the user to use the data for different purposes while detailed information on sampling time and statistical weighing in the different statistical analyses, tests, and figures will help to justify whether patterns or relations presented in figures or table hold or whether these patterns appear by chance.

Response: This is a dataset of multi-year average values, therefore, differences in spatial and temporal resolutions, and sample types should not have a big impact on the spatial distribution of these isotope values and their relationship with geographical and climatic factors. We therefore did not present the data separately based on these data differences.

Based on your suggestion, we supplemented the raw data we complied (see rawdata.xlsx). In addition, we added explanation in the manuscript that data were averaged at each site over time or depth to get the multi-year mean values.

As we explained it in the introduction, the values of such an averaged dataset are can be demonstrated in the following examples. First, it provides an observational basis for numeric simulations of the spatial distribution of snow isotopes across Antarctica, such as the mathematical methods used in Wang et al. (2009a, 2009b, 2010). Second, it can be used to reconstruct paleo-temperature and paleo-elevation changes (Werner et al., 2018). Finally, it can be used as a benchmark to evaluate isotope-enabled atmospheric general circulation models (AGCMs) and Rayleigh distillation isotope models (Markle and Steig, 2022; MD08; Steen-Larsen et al., 2016).

The dataset can be directly compared with model results, because it is a comparison of the annual or multi-year averages reflecting the modern climate state.

L32 Clarify what the difference between a data point and location.

Response: We deleted the data points without coordinates, and now there are 1698 data points. Not all locations have oxygen and deuterium isotopes, 1604 locations have oxygen isotopes, 1278 locations have deuterium isotopes. 1204 locations have both $\delta^{18}O$ and $\delta^2H$, from which d-excess (d-excess = $\delta^2H - 8 \times \delta^{18}O$) can be calculated. There is no essential difference between data points and locations.

L35 Specify already here geographic and climate information.

Response: We have specified geographic and climate information in the revised manuscript.

*"The dataset also contains geographic (elevation and distance from the nearest coast) and climate (temperature and snow accumulation) information."*

L52 Include reference to literature.

Response: We have added reference to literature.

*"covering only about 60 years (Wang et al., 2016)."*

L56 Clarify what "information and high fidelity" means.

Response: Ice core records have high fidelity because of low temperature environment and frozen state. Ice core records can well preserve the original climate information, while other records may not have such preservation conditions and the recorded climate information may be changed or distorted. We have interpreted the means of "information and high fidelity" in the revised manuscript.

*"Ice cores from Antarctica are critical archives for climate change due to their high resolution, long temporal coverage, large amount of high-quality climate and environmental information."*

L61 and L63 "spatial linear" remove spatial in L61 since the emphasis was on the core to avoid confusion with different cores located at various locations.

Response: We have removed the word "spatial".

*"Conventional temperature reconstructions by stable water isotopic records in ice cores rely on the empirical linear relationship between the isotopic composition in surface snow and ice and air temperature (Jouzel et al., 2003)."*

L62 and L62 Include references at the end of the sentence.

Response: We have added reference at the end of the sentence.

*"Conventional temperature reconstructions by stable water isotopic records in ice cores rely on the empirical linear relationship between the isotopic composition in surface snow and ice and air temperature (Jouzel et al., 2003)."*

L65 "uneven" Clarify what uneven means.

Response: We have clarified what uneven means in the revised manuscript. The sampling sites are mainly concentrated in low elevation areas and thus the spatial coverage of isotope data remains highly uneven.

*"Although isotopic observations have been recorded at over one thousand sampling sites, these sites are mainly concentrated in low elevation areas, leading to highly uneven spatial coverage of isotope data (MD08)."*

L70 "high-resolution" Specify resolution.

Response: We have specified the resolution. Resolution is 1 km.

*"using pure mathematical methods combined with 1 km high-resolution digital elevation models (Wang et al., 2009a, 2009b, 2010)."*

L74 Include reference to literature.

Response: We have added reference to literature.

*"Finally, the database can be used as a benchmark to evaluate isotope-enabled atmospheric general circulation models (AGCMs) and Rayleigh distillation isotope models (Markle and Steig, 2022; MD08; Steen-Larsen et al., 2016)."*

L77 "high altitude" not consistent use of altitude vs. elevation.

Response: We have used the word "elevation" to replace the word "altitude" throughout the manuscript.

*"Only a few data are available on the West Antarctic Ice Sheet and at high elevations of the East Antarctic interior regions (Figure 1a)."*

L78 Include a figure to highlight the distribution.

Response: We have included a figure to highlight the distribution.

*"Only a few data are available on the West Antarctic Ice Sheet and at high elevations of the interior East Antarctica (Figure 1a)."*

L79 "researchers", which ones?

Response: We have added the researcher's references.

*"After the pioneering work of MD08, numerous new samples and measurements have been acquired by different researchers (i.e. Ekaykin et al., 2020; Landais et al., 2017; Weinhart et al., 2021)."*

L89 Explain how the data was sourced.

Response: We have explained how the data was sourced. The data was sourced from literature and public data portals.

*"This article aims to provide some details on the collection of the isotopic measurements used to produce this updated Antarctic surface snow and ice isotopic dataset, including data sources (literature and public data portals), selection criteria, and spatial distribution."*

L91 "strong support" clarify who gave support, the funding agency or colleagues?

Response: We have clarified who gave support.

*"We received strong support from many authors and researchers in the process of data collection."*

L96 Give a brief introduction of "traverse sampling" not everyone might be familiar with this type of sampling approach. Include also other sampling approaches and sampling methods and devices used.

Response: We gave a brief introduction of "traverse sampling" in the manuscript. The traverse sampling of surface snow is usually carried out along an investigation route, often between Antarctic research stations. The usual sampling method is to dig snow pits or use metal instruments to carry out a certain depth of excavation. Our description in the revised draft is as follows.

*"The traverse sampling of surface snow is usually carried out along an investigation route, often between Antarctic research stations. The samples are usually collected from snow pits at certain depths (Dallmayr et al., 2021; Li et al., 2014; Weinhart et al., 2021)."*

L97 Specify who collected this data, the authors or different authors from the literature.

Response: Different authors from the literature collected this data. We have made changes in the revised manuscript.

*"Traverse sampling is an important way for Antarctic researchers (i.e. Qin et al., 1994; Weinhart et al., 2021) to obtain spatially distributed data of isotopic composition."*

L106 "we added two routes" Clarify whether own data or literature sourced data was used.

Response: We have clarified this sentence in the revised manuscript. This refers to the data from the literature.

*"For the high inland region of East Antarctica (90 to 120°E), we supplemented two routes of traverse data (Vostok−Dome B and Vostok−Dome C traverses; Ekaykin et al., 2020; Landais et al., 2017) into the original database."*

L108 "unreleased" Explain what unreleased mean here since the data seems to be published by Ekaykin et al. (2012). Please clarify.

Response: We removed the words "unreleased public".

*"Furthermore, in the area around the Vostok site, we included 89 snow pits data, such as the Vostok flow line data (Ekaykin et al., 2012)."*

L128 Explain what reliability and its context means.

Response: We changed this sentence as follows. "*As we aimed to compile a comprehensive, reasonable, and quality-controlled isotopic dataset of multi-year average values under modern climatic conditions, we excluded isotopic data with apparent seasonal bias (such as fresh surface snow) and those modified by post-depositional processes (such as surface glacial ice) at Dry Valleys*".

L129 Explain what a seasonal bias is.

Response: Surface fresh snow data may reflect seasonal values at the time of collection rather than the annual average signal. This is particularly true in coastal

areas, where the accumulation is very high and fresh snow has strong seasonal variations. A shallow surface snow sample could hence carry seasonal biases.

L132 Since the data was collected in the dry valley with little precipitation, the data could still be a multi-year average. Please clarify.

Response: Previous studies reported a high number of fresh surface snow and surface glacial samples with abnormal negative d-excess values (the range is about –10 to –5‰) in the Dry Valleys (Gooseff et al., 2006; Hu et al., 2022; MD08). Considering accumulation rates and the influences of post-deposition processes, we believe that it is unlikely that these fresh surface snow (5 cm) or surface glacial ice represent the multi-year average isotope values. Therefore, we excluded these surface samples from the Dry Valleys (Gooseff et al., 2006; MD08), and only retained data from snow pits (1 m).

L137 State clearly why only the last few decades were included and not all.

Response: Most of data come from modern samples, representing modern climate conditions. For consistency, we only included data of the last few decades from newly added ice cores, with the purpose of establishing average isotopic composition of Antarctic surface snow and ice under modern climatic conditions.

L146 Please also include additional geographic factors such as slope and aspect ...

Response: Since they are not significant factors affecting isotopic variations, slopes and aspects are not reported in any of the isotopic studies. Therefore, we did not include them in our dataset.

L157 Add reference to original authors.

Response: We added a reference.

*"the equilibrium and kinetic fractionation effects occurring during the formation of ice crystals at very low temperatures (Jouzel and Merlivat, 1984)."*

L196 Specify which correlation coefficient was used.

Response: The Pearson correlation coefficient was used in this study.

*"The Pearson's correlation coefficient between measured and observed $\delta^{18}O$ is slightly higher for ECHAM6-wiso (r = 0.97, model-data slope = 0.78, p < 0.01) than iCAM6 (r = 0.96, model-data slope = 0.84, p < 0.01) (Figs. 4a and 4b, Figs. 4e and 4f)."*

L223 Clarify how the comparison was performed (not described in text L190-onwards). It should be included in the method section.

Response: First, we average the model data over time. We then use linear interpolation to interpolate the model data to each sampling site. Then we performed a regression analysis between the measured and modeled data. We added more explanation in the manuscript.

*"In order to compare the observed and modeled data, we averaged the model data over time. We then interpolate the model data to the sampling site using linear interpolation method. Next, we performed regression analysis between the measured and modeled data. For more details on model evaluation methods, please refer to the supplementary material of Werner et al. (2018)."*

From L190, the model was run from 1979-2018, but the data had different temporal resolutions. Is it fair to compare these data, and what can we learn from this non-equal comparison?! Please clarify and discuss.

Response: The dataset contains multi-year averaged isotopic values, although the time span of the sampling points is different. Even if the model period (i.e., 1979–2018) and the dataset period (~1905–2020) are not exactly the same, we do not expect large changes when comparing the climatological means. Such comparisons are frequently performed to evaluate the skills of the models to simulate the spatial distribution of $\delta^{18}O$ and $\delta D$ in Antarctica (e.g., Risi et al., 2010; Werner et al., 2011, 2018). Such comparison is especially useful to evaluate the modeling of d-excess of Antarctic precipitation, which is very sensitive to the cloud microphysics of the models and the parameterization of the supersaturation (Cauquoin and Werner, 2021; Dütsch et al., 2019; Risi et al., 2010). Although this comparison has some shortcomings, it is reasonable in the average state of climate. We have clarified and discussed this in the manuscript.

*"Even if the model period (i.e., 1979–2018) and the dataset period (~1905–2020) are not exactly the same, we do not expect large changes when comparing the climatological means. Such comparisons are frequently performed to evaluate the skills of the models to simulate the spatial distribution of $\delta^{18}O$ and $\delta D$ in Antarctica (e.g., Risi et al., 2010; Werner et al., 2011, 2018). Such comparison is especially useful to evaluate the modeling of d-excess of Antarctic precipitation, which is very sensitive to the cloud microphysics of the models and the parameterization of the supersaturation (Cauquoin and Werner, 2021; Dütsch et al., 2019; Risi et al., 2010). Although this comparison has some shortcomings, it is reasonable in the average state of climate."*

Table 1 Explain what are sufficient number of measurements are and why and based on what the threshold of 10 was chosen.

Response: The threshold of 10 is used by Masson-Delmotte et al. (2008). So, we used the same threshold to be consistent with the original data. Based on this criterion, sufficient number of measurements means 10 or more measurements at each sample location, which are used to establish the average values of the isotopic composition of surface snow and ice.

Figure 1 c choose different symbols similar to fig 4 for data from various data sources.

Response: Based on your suggestion, we used different colors to differentiate various data sources. Following is the revised Fig. 1.

[Figure]

**Figure 1.** The comprehensive dataset of Antarctic surface snow and ice isotopic observations. The black points indicate the original dataset of MD08 (a), and the blue points represent our newly added points (b), and the black and blue points indicate all data locations (c). Major new sampling traverses of isotopic observations in Antarctica (d).

Figure 4 Rearrange panels with column $^{18}$O and one column D-excess. Include different symbols and colors indicating different time scales.

Response: Our data has only a time scale, the annual average. We think the different time scales you are talking about in this question refer to different time spans. We rearranged the order of subplots as suggested. However, since we have so much data,

the density of the dots does not allow clear differentiations of different symbols. So, we only changed the color, not the symbol. The current re-draw figure is as follows.

[Figure]

**Figure 3.** Scatter plots for (a) δ¹⁸O vs. distance to the nearest coast and (b) δ¹⁸O vs. elevation. (c) and (d), same as (a) and (b), but for d-excess. (e) Scatter plot for δ¹⁸O vs. temperature. (f), same as (e), but for d-excess. The open white circles indicate the original dataset of MD08, and the black fill points represent our newly added points. Among them, color dots represent the MD08 database at different time spans, and gray dots represent newly added points with different time spans.

Figure 5 is hard to read too much data and color scheme, and symbols.

Response: We revised Fig. 5 based on your feedback. The revised figure is as follows.

[Figure]

**Figure 4.** Comparisons of observed water isotopes data with the ECHAM6-wiso and iCAM6 simulations. (a) The observed $\delta^{18}O$ (filled circles) and the simulated $\delta^{18}O$ (background color ramp) by ECHAM6-wiso model. (b) Scatter plot of the simulated $\delta^{18}O$ (y-axis) vs. observational $\delta^{18}O$ (x-axis). (c) and (d), same as (a) and (b), but for d-excess. (e) and (f), same as (a) and (b), but for iCAM6. (g) and (h), same as (e) and (f), but for d-excess. Letter r represents the correlation coefficient, slope represents the slope of linear regression, and rmse represents the root mean square error.